# Generating context-specific sports training plans by combining generative adversarial networks

Juquan Tan[1], Jingwen Chen [2]*

1 College of P.E.Teaching, South China Agricultural University, Guangzhou, Guangdong, China, 2 College of Education for the Future, Beijing Normal University, Zhuhai, Guangdong, China

* jingwen@bnu.edu.cn

**Data Availability Statement:** All relevant data are within the manuscript.

**Funding:** The author(s) received no specific funding for this work.

## Abstract

Personalized sports training plans are essential for addressing individual athlete needs, but traditional methods often need to integrate diverse data types, limiting adaptability and effectiveness. Existing machine learning (ML) and rule-based approaches cannot dynamically generate context-specific training programs, reducing their applicability in real-world scenarios. This study aims to develop a Generative Adversarial Network (GAN)- based framework to create context-specific training plans by integrating numeric attributes (e.g., age, heart rate) and motion features from video data. The research focuses on improving context-specific efficiency and real-time adaptability while addressing the limitations of traditional methods. The proposed GAN framework combines numeric and motion features using a generator-discriminator architecture to produce tailored training plans. The model is evaluated quantitatively through metrics like mean square error (MSE) and generation time and qualitatively through subjective ratings from athletes and coaches using a five-point Likert scale for context-specific, scientificity, applicability, and feasibility. Statistical significance is analyzed using ANOVA testing. The proposed GAN model outperforms traditional ML and rule-based methods, achieving a 22% reduction in MSE and a 45% improvement in generation time. Subjective evaluations reveal significant improvements in context-specific and applicability, with ratings averaging 4.8/5 compared to 3.9/5 for baseline models. The GAN framework effectively integrates multimodal data, demonstrating dynamic adaptability and high efficiency suitable for real-world applications. The proposed GAN-based framework advances the generation of personalized sports training plans by integrating numeric and motion data, achieving superior adaptability and efficiency. These results highlight the model's potential for practical deployment in athletic coaching systems, addressing critical gaps in existing methodologies and offering scalable solutions for individualized training.

## 1. Introduction

In cultivating excellent athletes, sports training is very important [1]. With the rapid development of technology and the continuous improvement of sports competition level, enhancing

**Competing interests:** The authors have declared that no competing interests exist.

the effectiveness of sports training through modern technology has become an important means to improve the competitive level of athletes [2, 3]. The human body's different growth and development patterns result in differences among individuals at various project stages. Traditional sports training methods often rely on the coach's experience to provide suggestions [4], lacking personalized, scientific [5–7], and targeted guidance for students [8]. This not only affects the training effectiveness of athletes but may also lead to resource waste and physical injuries to athletes. Therefore, exploring a method to generate personalized sports training plans is essential. This article combines GAN technology to develop customized exercise plans, providing athletes with more personalized and effective training plans to improve their effectiveness and competitiveness.

Through reviewing relevant literature, it has been found that in recent years, many scholars have explored personalized generation of sports training programs by using different techniques [9, 10]. Shin et al. [11] used a large language model to develop personalized exercise plans. Although this method can assist in generating customized training plans, it ignores athletes' real-time feedback and dynamic changes, resulting in a lack of real-time flexibility in the training plan. Cao et al. [12] used ML-based image processing techniques to provide personalized training for football players. Although this method improves the context-specific training plan to a certain extent, problems exist, such as weak model generalization ability and lack of diversity. Li and Shi [13] applied computer Internet of Things technology in gymnastics teaching and training, which can provide personalized training suggestions. The application of IoT technology requires corresponding equipment and infrastructure support, relying too heavily on specific devices and platforms.

To overcome the problems of the above methods, some researchers have attempted to apply GAN technology to generate personalized exercise training plans. GAN is a deep-learning architecture [14, 15]. Compared with traditional methods, GAN-based technology can generate high-quality and diverse samples and the advantages of real-time personalized scheme generation. Researchers have also studied GAN technology in data generation and personalized recommendation fields. Wang et al. [16] used conditional GAN to solve the problem of incomplete characterization of personalized features in human gait generation. Cao et al. [17] trains personalized models for customers through GAN to meet their needs better. Wen et al. [18] generated a personalized recommendation framework based on conditional GAN. Yoon et al. [19] Yuan et al. used the GAN framework to generate synthetic data, minimizing patient recognizability, to achieve more accurate decision-making and personalized treatment. [20] applied GAN to personalized sentence generation, which combines commonly used function words and content words as input features of the generator and generates personalized sentences by discriminating constraint conditions through a discriminator. Ali et al. [21] utilized GAN to generate a recommendation model that helps users provide personalized citations. Shi and Luo [22] applied Conditional Generative Adversarial Nets (CGAN) to personalized clothing recommendation and generation. Sun et al. [23] used GAN to provide users with personalized paper resources, generating resources that match user interests and preferences by using semantic features of papers as input features. Based on GAN, Gao et al. [24] model users' long-term stable and recent dynamic preferences through a game of generator and discriminator, providing personalized user recommendations. Gao et al. [25] used an adversarial network consisting of a generator and discriminator to generate personalized travel suggestions and demonstrated the effectiveness and efficiency of the proposed model through experiments. Traditional sports training methods and algorithms often fail to address the unique needs of individual athletes, relying on static, generalized plans that lack dynamic adaptability and real-time customization. Current methods do not handle multiple sources of information, including numeric attributes of an athlete and motion characteristics used to

design individualized programs appropriately. This limitation lowers the training efficiency and results that apply to context and sub-optimum. This research employs GAN technology to create contextualized sports training schedules through combinational capabilities of numeric and motion data. The scope includes.

- Developing a GAN-based framework capable of fusing multimodal inputs to create personalized plans.

- Evaluating the model's performance against traditional ML and rule-based approaches using quantitative and subjective metrics.

- Exploring scalability and practical implications for real-world applications in sports coaching systems.

This study proposes a new GAN architecture for creating explanatory sports training plans using numeric attributes interacting with motion features derived from videos. The dynamic of the adversarial training of the model also guarantees personalized and high-quality responses, making the approach superior to the traditional efficiency, context specificity, and applicability approaches. Qualitative self-assertions of the athletes and coaches supported by significance tests verify that the model delivers answers to multiple training requirements. The GAN's GPU-accelerated architecture confirms real-time applicability, thereby underlining its relevance for current athletic training systems.

Based on prior inadequacies of traditional training plans in terms of context-specific execution and scientificity, this study proposes to use GAN to generate personalized sports training plans. Also, multidimensional primary data, like physical fitness data of athletes and training performance data, can synthesize a GAN model that is effective in sports training. It can be feasible to produce training plans according to the characteristics of athletes, and comparative tests can test the validity of this model. The research methods and results of this article can offer athletes more scientific and individualized training information and encourage new advances in training.

The paper is organized as follows. Section 2 reviews related work and identifies research gaps. Section 3 outlines the methodology, including data collection and the GAN framework. Section 4 presents experimental results and analyses. Section 5 concludes with key findings, limitations, and future research directions.

## 2. Literature review

Researchers have recently explored various methods to design effective and personalized sports training plans. While traditional approaches have focused on generic frameworks, emerging techniques like artificial intelligence (AI) and ML have introduced new possibilities for context-specific [26, 27]. However, these technologies still face challenges related to adaptability, integration of multimodal data, and real-world applicability. This section critically reviews existing approaches and highlights their limitations and positions in this proposed study within this context.

### 2.1 Traditional approaches to sports training plans

Traditional training prescriptions are delivered based on the predetermined algorism and coaches' professional experience, pay more attention to the group average, and lack consideration of individual differences. For example, Zhang and Hou [4] used the video image processing technique to promote sports action recognition, improving the training plan in general conditions but without considering the individual differences of athletes. Likewise, Pickering

and Kiely [28] stressed the relevance of individualized training translucent schemes, but they did not integrate adaptive changes according to time-bound feedback or individual differences. These methods show the problems with relying on Taylor-like or coach-dependent models that do not consider the differences in individual athletes.

## 2.2 Machine learning-based methods

Applying ML has helped the context-specific thrive of sports training through data models. Shin et al. [11] developed exercise plans for users with the help of large language models (LLMs); the result was not the complete individualization of exercise plans but a certain attempt. Element, however, was rigid in that it offered no mechanism for changing the planned training in case this was needed due to changes in the performance of athletes or their physical conditions. Cao et al. [12] used image processing with the help of ML to propose strength training programs for soccer players. These methods are beneficial for increasing specific results but have low transferability to other kinds of sports and training. Alas, the current state of affairs and an array of strengths associated with ML methods make specific issues apparent: the inadmissibility of integrating various kinds of data and dynamic structural changes.

## 2.3 Integration of IoT and AI in sports training

Issues with real-time monitoring through IoT-enabled systems have been of the essence in sports training, especially in training sessions. In their work, Li and Shi [13] described the application of IoT technology to improve gymnastic training by providing patient-specific advice based on sensor data. However, these systems were designed to work off special-purpose circuits and were not scalable or easily accessible. In the same year, Ghanvatkar et al. [9] suggested AI-based physical activity interventions concerning IoT data. However, they observed the barriers of the approach, which limit the interventions' flexibility for individual athletes and their indefinite sustainability. As a result, albeit as systems for providing feedback from various processes and events, IoT programs are limited in applicability because they depend on outside devices and equipment.

## 2.4 GAN-based techniques

Generative adversarial networks (GANs) have limited themselves to generating high-fidelity, customized content in rich domains. Khan et al. employed conditional GANs. [29] in modeling human gait, thereby achieving the modeling of individual traits. Most research on GANs has been for feature extraction. Recently, Yoon et al. [19], GANs were used to synthesize pretend health data for personalized decisions, showing their promise for data synthesis and versatility. However, little research has been done on the application of GANs in the training of athletes. Existing studies have not fully integrated diverse data modalities, such as numeric attributes and video-derived features, into a single framework for generating training plans.

## 2.5 Research gap and contribution

The following gaps in the existing literature highlight the need for this proposed study. Current models struggle to cohesively fuse numeric attributes (e.g., age, weight, heart rate) with video-based motion features. Few methods incorporate real-time feedback to adjust training plans dynamically. Limited case studies or examples demonstrate the real-world application of these approaches.

These gaps are addressed in this proposed study using GANs to produce approximations of context-specific training plans. Numeric attributes combined with motion features learned

from video data through feature extraction by a pre-trained Convolutional Neural Network (CNN). The static model presented herein is vulnerable to the prepared athlete-formula-generated profile and outperforms the traditional and ML models in generating more efficient and accurate individualized plans. The effectiveness of this approach is supported systematically by evaluating the mean square error (MSE), generation speed, and qualitative remarks from the athletes themselves, showing the practicality of the approach. By closing such gaps in this proposed work, there will be tremendous progress in creating program-specific training for individual athletes.

## 3. Generation of personalized sports training plans based on GAN

This section presents the methodology for generating personalized sports training plans using a GAN. The proposed framework integrates numeric attributes (e.g., age, weight, heart rate) and motion features extracted from video data to create a multimodal input representation. The generator produces context-specific training plans tailored to individual athlete profiles, while the discriminator evaluates the authenticity and relevance of the generated plans. Adversarial training enables iterative refinement, ensuring high-quality outputs. Fig 1 shows the proposed methodology.

### 3.1 Data collection and preprocessing

The recruitment period for this study began on 01/04/2024 and ended on 01/05/2024. In the data collection stage, this article first collected individual athletes' sports training videos and personal feature data using the Olympic Sports Dataset dataset. Personal characteristic data includes age, height, weight, heart rate, etc. The data used in this study consists of a collection of high-resolution video recordings capturing various training activities performed by athletes. Each video sample was recorded at a resolution of 1920x1080 pixels with a frame rate of 30 frames per second (FPS). These specifications were selected to ensure sufficient detail for

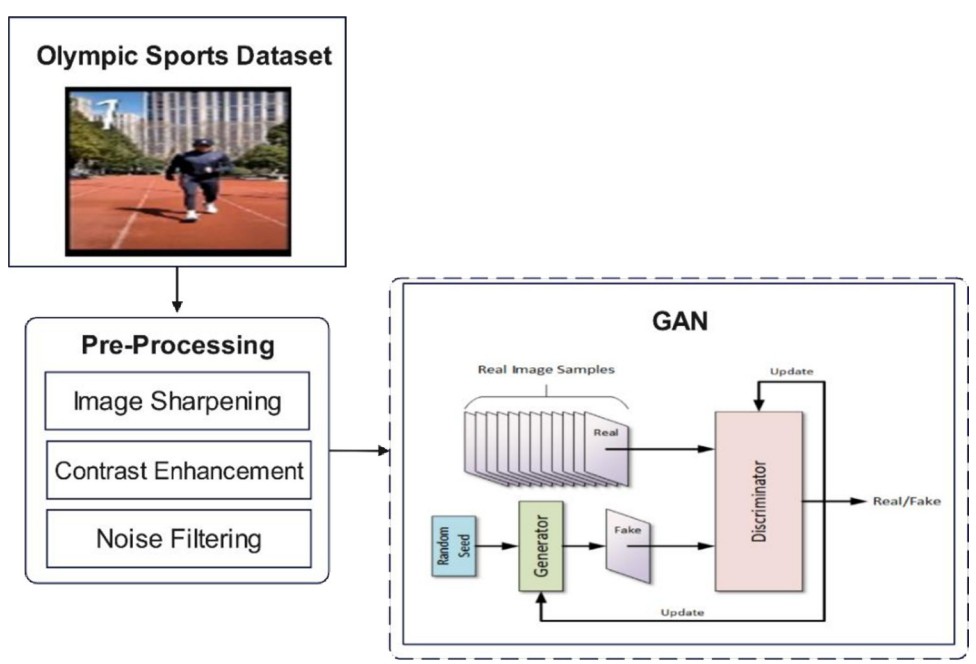

**Fig 1. Flowchart of the proposed research.**

**Table 1. Key attributes and characteristics of the dataset for generating personalized sports training plans.**

| Attribute | Details |
|---|---|
| Video Resolution | 1920x1080 pixels |
| Frame Rate | 30 FPS |
| Average Video Length | 10 minutes (approx. 18,000 frames) |
| Total Videos | 1,000 |
| Number of Athletes | 100 |
| Devices Used | GoPro HERO9 Black, Garmin Forerunner 945 |
| Metadata | Age, weight, heart rate, performance goals |

motion analysis while maintaining compatibility with standard video processing tools. On average, each video is approximately 10 minutes in duration, resulting in about 18,000 frames per video. For preprocessing purposes, keyframes were extracted at intervals of 0.5 seconds, yielding approximately 1,200 frames per video. The dataset consists of 1,000 videos collected from 100 athletes, each contributing approximately 10 video samples. These videos capture diverse training activities, including aerobic exercises, interval training, stretching, and technical drills. This variety ensures that the dataset represents a broad spectrum of training scenarios. To maintain uniformity, all videos were recorded using GoPro HERO9 Black cameras. These cameras were chosen for their high-definition recording capabilities and stability features, crucial for capturing fast-moving athletes. The cameras were mounted on tripods at fixed positions and aligned to provide consistent angles and complete visibility of the athletes. Sports training video data requires a series of preprocessing operations such as video frame extraction, image preprocessing, and motion posture annotation to analyze information such as athlete's motion posture, intensity, and frequency during the training process. Table 1 provides a comprehensive overview of the dataset employed in the study.

**3.1.1 Video frame extraction.** When preprocessing motion training videos, the first step is to perform video frame extraction on the collected motion training videos, which converts the videos into continuous static images for subsequent image processing and analysis. Here, the image processing tool OpenCV can extract frames from the video. Firstly, it can use the cv2. VideoCapture() function to open the video file and pass in the path of the video file. Then, the video. read() method can be used to loop through each frame of the video. For each frame of the image, it can be named using the frame counter frame_count and saved using the cv2. imwrite() function. Fig 2 shows the effect of partially extracting video frames.

**3.1.2 Image preprocessing.** After performing frame extraction on motion videos, the next step is to preprocess the video frame images, including image sharpening, contrast enhancement, and noise filtering. The purpose of these operations is to eliminate noise in the image, enhance the features of the image, and improve the quality and clarity of the image, providing more accurate image data for subsequent motion pose annotation. Firstly, the image can be sharpened to enhance its high-frequency details and make it clearer and more vivid. In this study, Laplacian filters can be used to achieve sharpening processing. Laplace filter is a commonly used edge detection filter that highlights high-frequency details in an image by performing second-order differentiation on the image [30]. The processing process is shown in Eq (1).

$$Sharpened(x, y) = Input(x, y) + k*(Input(x, y) - Laplacian\_Filtered(x, y)) \qquad (1)$$

In this Eq, $Sharpened(x,y)$ represents the pixel value in the sharpened image, and $Input(x,y)$ is the original image value. $k$ is the sharpening parameter, and $Laplacian\_Filtered(x,y)$ is the pixel value in the original image processed by a Laplace filter. After the sharpening process is

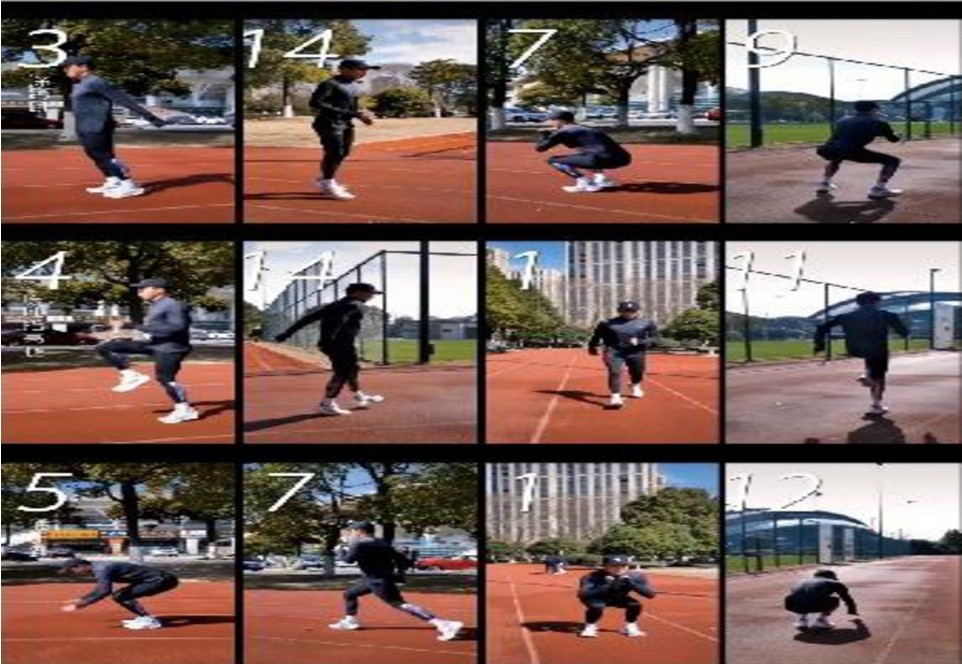

**Fig 2. Video frame extraction effect.**

completed, it is necessary to perform contrast enhancement on the image. Contrast enhancement is achieved by adjusting the brightness distribution of the image to enhance its contrast. Histogram equalization is used here to achieve contrast enhancement. Histogram equalization expands the dynamic range of the image by reallocating pixel values by calculating the cumulative distribution function (CDF) of the image [31]. The implementation process is shown in Eq (2).

$$Enhanced(x, y) = \frac{CDF(Input(x, y)) - \min\_CDF}{\max\_CDF - \min\_CDF} \times (\max\_pixel\_value - \min\_pixel\_value) + \min\_pixel\_value) \quad (2)$$

Here, $Enhanced(x,y)$ represents the pixel value of the image after contrast enhancement and $CDF(Input(x,y))$ represents the cumulative distribution function of the input image. $min\_CDF$ and $max\_CDF$ are the minimum and maximum cumulative distribution values of the input image, respectively. $min\_pixel\_value$ and $max\_pixel\_value$ are the output image's minimum and maximum pixel values, respectively. After equalization through this histogram, the details and contrast in the image can be enhanced, making the image clearer and fuller. Finally, noise filtering can be applied to the image to reduce noise interference and improve the quality and clarity of the image. Here, noise filtering can be achieved using a Gaussian Filter, which is a linear smoothing filter that reduces noise by applying spatial Gaussian smoothing to the image [32]. It uses a Gaussian function to weigh the average of the pixels in the image, suppressing noise. The processing process is shown in Eq (3).

$$Filtered(x, y) = \sum_{i=-k}^{k} \sum_{j=-k}^{k} \frac{1}{2\pi\sigma^2} e^{-\frac{i^2+j^2}{2\sigma^2}} \times Input(x + i, y + i) \quad (3)$$

Among them, $Filtered(x,y)$ represents the pixel value of the image after noise filtering; $\sigma$ is the standard deviation of the Gaussian kernel, and $Input(x+i,y+i)$ represents the pixel value of

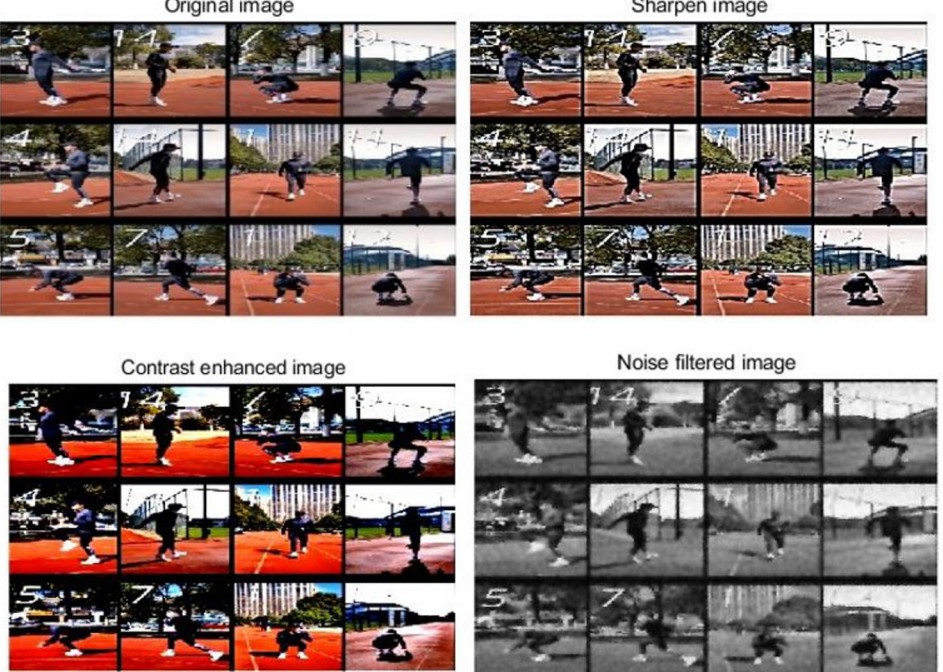

**Fig 3. Image preprocessing effect.**

the input image at the position (*x+i,y+i*). After these image preprocessing operations, the image is shown in Fig 3.

**3.1.3 Sports posture annotation.** In the stage of sports posture annotation, this article adopts a method based on human key point detection to locate key points in the preprocessed image and annotate the athlete's posture information for subsequent analysis. This method automatically detects the key point positions of athletes in images through an OpenPose model based on deep learning. It uses a deep-CNN to extract features from the image and predicts the position of key points through a regression network. The mathematical expression is shown in Eq (4).

$$Heatmap(x, y, c) = exp\left(-\frac{(x - x_c)^2 + (y - y_c)^2}{2\sigma^2}\right)$$ (4)

Here, *Heatmap(x,y,c)* represents the heatmap of key point positions; $\sigma$ is the standard deviation of the Gaussian kernel, and *c* is the category of key points. Accurate key point positions can be obtained by training the model and annotating the athlete's posture information.

The results of motion posture annotation are shown in Fig 4. By annotating the postures of athletes in training videos, it can analyze the habitual posture and action information of athletes during the training process, preparing for personalized program recommendations in the future.

## 3.2 GAN model architecture design

This article designs a GAN-based model architecture to achieve personalized motion training scheme generation. Among them, the GAN system includes two deep neural networks—the generator network and the discriminator network [32, 33]. These two networks train models in adversarial games. One of the networks attempted to generate new data, and another tried

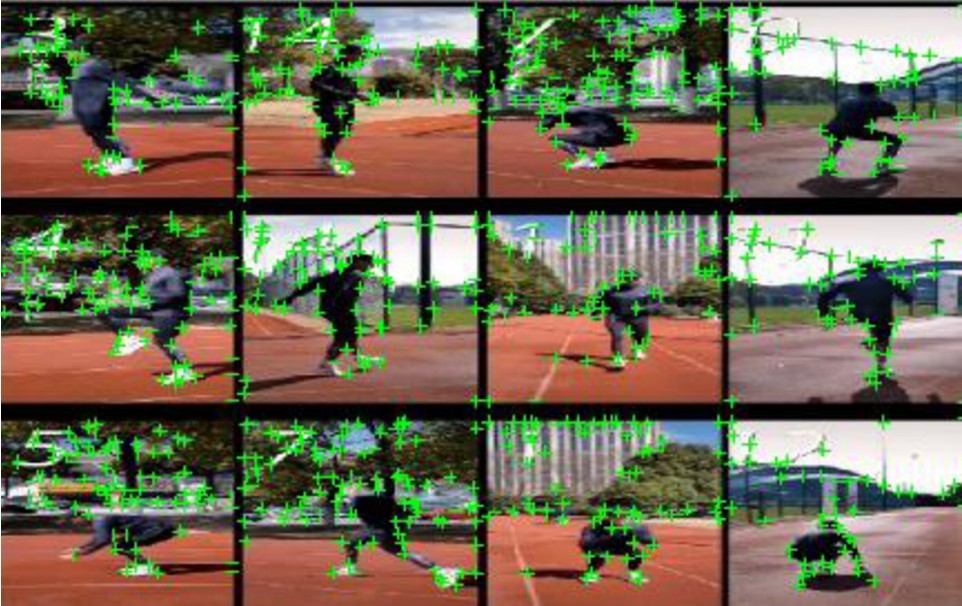

**Fig 4. Sports posture annotation diagram.**

to predict whether the output was fake or real data, so the model architecture consists of a generator and a discriminator. The task of the generator is to generate personalized training plans [34], and the discriminator's task is to evaluate the authenticity of the generated plans. The numeric attributes (e.g., age, height, weight, heart rate) were first encoded into feature vectors, while motion features were extracted from video frames using a pre-trained CNN. These two modalities were then concatenated to form a unified input representation, which was used by the GAN generator to create context-specific training plans. The model architecture diagram is shown in Fig 5.

The generator adopts a deep neural network (DNN) as its infrastructure. It consists of multiple hidden layers and activation functions, used to learn and generate personalized motion training schemes, and uses a multi-layer fully connected network to build the generator. The task of the generator is to map the input individual athlete's feature data $x$ to the training scheme space and output a personalized sports training scheme $G(x)$, whose mathematical expression is shown in Eq (5).

$$G(x) = \mathrm{ReLU}(W_{out} \cdot \mathrm{ReLU}(W_{h2} \cdot \mathrm{ReLU}(W_{h1} \cdot x + b_{h1}) + b_{h2}) + b_{out}) \tag{5}$$

Among them, $x$ is the input individual athlete feature data; $W_{h1}$ and $b_{h1}$ are the weights and biases of the first hidden layer; $W_{h2}$ and $b_{h2}$ are the weights and biases of the second hidden layer; $W_{out}$ and $b_{out}$ are the weights and biases of the output layer; and ReLU represents the modified linear unit function.

The discriminator, as another part of the GAN model, is another deep neural network used to evaluate the authenticity of the generated schemes. The discriminator receives the scheme generated by the generator and the real scheme as inputs and outputs a probability value between 0 and 1, indicating the probability that the input scheme is the real scheme. Its mathematical expression is shown in Eq (6).

$$D(x) = \sigma(W_d x + b_d) \tag{6}$$

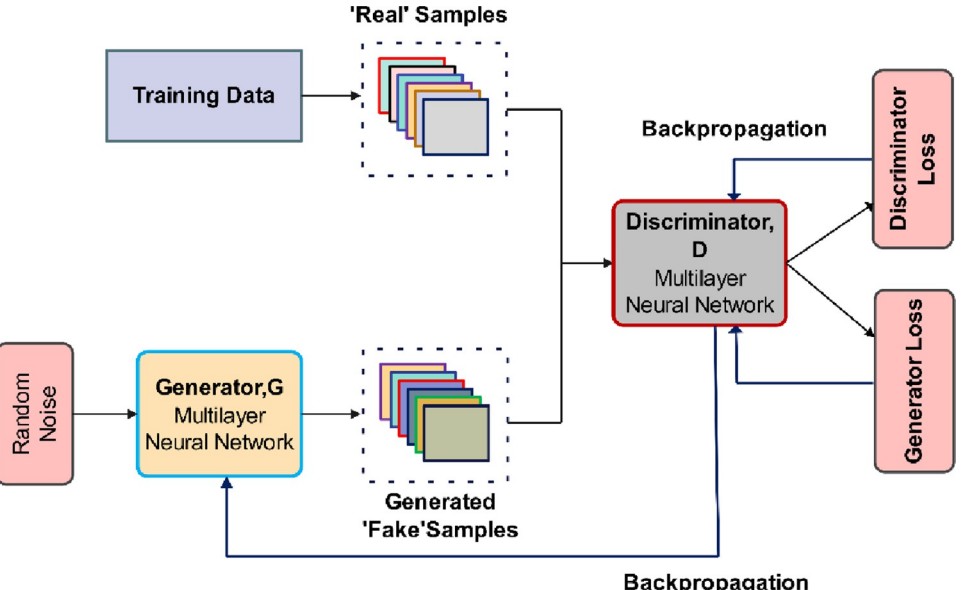

**Fig 5. GAN model architecture diagram.**

Here, $D(x)$ represents the output discrimination result; $\sigma$ is the activation function; $W_d$ is the convolution kernel, and $b_d$ is the bias term. By continuously alternating the training of the generator and discriminator, the generator can ultimately generate high-quality and personalized motion training plans. In contrast, the discriminator can accurately distinguish between real samples and generated samples.

## 3.3 Defining loss functions and optimization strategies

In this study, for GAN to effectively generate personalized motion training plans, it is necessary to define appropriate loss functions and optimization strategies. The loss function is used to measure the performance of the generator and discriminator, while optimization strategies are used to update network parameters to minimize the loss function iteratively. Firstly, the loss function of the generator can be defined. To make the generated personalized training scheme as close as possible to the real scheme, the average log-likelihood loss function was used as the loss function of the generator to measure the difference between the generated scheme and the real scheme. The implementation process is shown in Eq (7).

$$L_G = -\frac{1}{N}\sum_{i=1}^{N} log(D(G(x_i))) \tag{7}$$

In this Eq, $N$ represents the number of training samples; $x_i$ represents the input individual motion data; $G(x_i)$ represents the personalized training plan generated by the generator on the input data $x_i$, and $D()$ represents the discriminator. The generator's goal is to minimize the loss function to improve the authenticity of the generated scheme. Next, the loss function of the discriminator can be defined to distinguish accurately between the generated scheme and the actual scheme. To achieve this goal, the cross entropy loss function can be used as the loss function of the discriminator, as represented by Eq (8).

$$L_D = -\frac{1}{N}\sum_{i=1}^{N}(log(D(x_i)) + log(1 - D(G(x_i)))) \tag{8}$$

Here, $D(G(x_i))$ represents the evaluation result of the discriminator on the generation scheme $G(x_i)$. The discriminator aims to minimize the loss function, enabling it to accurately distinguish between generated and real schemes. Thus, the generator generates more realistic personalized training schemes.

Regarding optimization strategy, the Stochastic Gradient Descent (SGD) algorithm can be used to iteratively update the parameters of the generator and discriminator to optimize the parameters in GAN. The generator updates the parameters based on the negative gradient of its loss function, and the implementation process is shown in Eq (9).

$$\theta_G = \theta_G - \eta \cdot \nabla_{\theta_G} L_G \tag{9}$$

In this Eq, $\theta_G$ represents the parameters of the generator; $\eta$ is the learning rate, and $\nabla_{\theta_G} L_G$ is the gradient of the generator loss function concerning the parameters. Using the backpropagation algorithm, gradients can be calculated, and the learning rate can be used to control the update speed of parameters. For the discriminator, the parameters are updated based on the negative gradient of the loss function of the discriminator, as shown in Eq (10).

$$\theta_D = \theta_D - \eta \cdot \nabla_{\theta_D} L_D \tag{10}$$

Here, $\theta_D$ represents the parameters of the discriminator and $\nabla_{\theta_D} L_D$ is the gradient of the discriminator loss function concerning the parameters. By defining the loss function above and implementing optimization strategies, the parameters of the generator and discriminator can be iteratively updated, gradually improving the quality and context-specific of the generated plan, thereby generating personalized motion training plans.

### 3.4 Conduct adversarial training

Adversarial training is the core part of GAN, aimed at improving the quality and context-specific generated training schemes through the game process between the generator and discriminator. In generating personalized solutions through adversarial training, the preprocessed dataset is first used for model training. This includes inputting individual feature data into the generator, generating personalized motion training plans, and simultaneously inputting the generated and real plans into the discriminator for adversarial learning. In the adversarial training process, the goal is to alternate the maximum and minimum adversarial loss functions, as shown in Eq (11), to optimize the parameters of the generator and discriminator.

$$min_G \; max_D \; V(D, G) = \text{IE}_{x \sim p_{data}(x)}[log \, D\,(x)] + \text{IE}_{z \sim p(z)}[\log(1 - D(G(z)))] \tag{11}$$

Among them, $G$ represents the generator; $D$ represents the discriminator; IE is the expected value; $p_{date}(x)$ represents the distribution of real data; $z$ represents the input noise vector of the generator and $p(z)$ represents the noise distribution. Through this iterative adversarial training process, the generator and discriminator compete and adjust with each other, ultimately achieving a dynamic equilibrium. The generator generates personalized training plans closer to the real plan by deceiving the discriminator. In contrast, the discriminator improves its discriminative ability by accurately distinguishing between the generated and real plans. In this way, the generated sports training plan can better meet the special needs of individual athletes and improve training effectiveness and sports performance levels.

## 4. Model evaluation and result

### 4.1 Experimental setup

**4.1.1 Experimental environment configuration.** This experiment used a server equipped with an NVIDIA GeForce RTX 3090 GPU, Intel Core i9-11900K CPU, and 32GB memory to implement the GAN model using the Python programming language and the deep learning framework TensorFlow [35].

**4.1.2 Dataset selection.** To evaluate the personalized sports training scheme generated by GAN, this experiment selected the Sports Performance Dataset from the UCI ML repository as the experimental dataset. This dataset contains many athletes' sports data and related features, which can be used to train and evaluate the model in this paper. Data from 100 athletes can be selected as experimental samples from the dataset, and representative athletes are selected for the experiment based on their different characteristics and needs.

**4.1.3 Athlete sample selection.** A total of 100 athletes were selected to evaluate the proposed GAN-based model. The selection ensured diversity in attributes, data quality, and training goals. Athletes represented a range of ages (18–45 years), weights (55–90 kg), and heights (155–195 cm), with an average heart rate of 135.4 bpm. Training objectives included endurance (40%), speed (35%), and flexibility (25%), as shown in Table 2.

This can ensure that the model is evaluated on different types of athletes to verify its effectiveness in generating personalized plans.

### 4.2 Experimental process

This experiment compares and analyzes the GAN model with three traditional algorithm models (ML-based, rule-based, and statistical based) to evaluate the performance of combining GAN to generate personalized motion training schemes. In terms of experimental setup, the dataset can be divided into training and testing sets, using 30% of the data as the training set and 70% as the testing set. Next, for each model, this experiment objectively calculated the evaluation indicators for each model's generation speed, MSE value, and response speed when generating personalized solutions. In addition, it also conducted subjective evaluation comparisons and collected feedback from these 100 athletes on personalized plans generated by different methods. A questionnaire survey can gather input from athletes on their preferences, feasibility, adaptability, and other aspects of the plan. Through comprehensive analysis of experimental results and subjective evaluation comparison, the performance of GAN and traditional algorithm models can be comprehensively evaluated and compared to draw conclusions.

### 4.3 Experimental results

**(1) Display of personalized sports training plans generated by combining GAN.** Firstly, to verify the feasibility of combining GAN to generate personalized sports training plans in this article, a running athlete can be used as an example. After inputting the athlete's

**Table 2. Summary of athlete characteristics.**

| Attribute | Range/Values | Average |
|---|---|---|
| Age (years) | 18–45 | 29.3 |
| Weight (kg) | 55–90 | 72.5 |
| Height (cm) | 155–195 | 174.2 |
| Heart Rate (bpm) | 55–190 | 135.4 |

**Table 3. Example of a personalized sports training program.**

| Date | Training program | Duration (hours) | Strength |
|---|---|---|---|
| Monday | Aerobic running | 1.5 | Medium |
| | Core training | 0.5 | High |
| Tuesday | Interval training | 1.0 | High |
| | Stretching training | 0.5 | Low |
| Wednesday | Aerobic running | 1.5 | Medium |
| | Technical exercises | 1.0 | Medium |
| Thursday | Interval training | 1.0 | High |
| | Core training | 0.5 | High |
| Friday | Aerobic running | 1.5 | Medium |
| | Stretching training | 0.5 | Low |
| Saturday | Long distance running | 2.0 | Medium |
| | Technical exercises | 1.0 | Medium |
| Sunday | Rest | | |

characteristic data into the GAN model, taking into account the runner's physical fitness, technical level, and training objectives, a personalized training plan suitable for this athlete is generated. The training plan includes daily and weekly training, including physical fitness training, technical training, and specialized training, as well as rest time. An example of the generated plan is shown in Table 3.

**(2) The variation curve of the loss function during the training iteration process.** In this experiment, the convergence of the GAN model was observed by plotting the changes in the loss function curve during the training process. The loss function curve includes two parts: generator loss and discriminator loss, where the generator loss reflects the performance of the generator network in generating personalized motion training schemes. The discriminator loss reflects the discriminator network's performance in evaluating the generated scheme's authenticity. This experiment records the unchanged loss function values of the generator and discriminator over 100 training iterations, as shown in Fig 6. Among them, the horizontal axis represents the number of training iterations, and the vertical axis represents the loss value. The blue curve represents the change in generator loss, while the red curve represents the change in discriminator loss. The lower the loss value of the generator, the closer the sample generated by the generator is to the real sample.

The lower the loss value of the discriminator, the stronger its ability to distinguish between real and generated samples. As the number of training iterations increases, the loss value of the generator gradually decreases from around 0.8 and tends to stabilize. This indicates that the generator network gradually improves the quality of generated samples during the learning process, making them closer to real samples. The loss value of the discriminator gradually decreases from about 0.6 and fluctuates within a certain range, eventually stabilizing. This indicates that the discriminator network has gradually improved its ability to distinguish between real and generated samples during the learning process and can more accurately determine the source of samples. The GAN model can effectively generate high-quality motion training schemes through the game process between the generator and discriminator. The discriminator can accurately distinguish between generated and real samples, which strongly supports the design and application of personalized sports training programs.

To comprehensively evaluate the proposed GAN model, the performance assessed using standard metrics, including accuracy, precision, recall, F1-Measure, and confusion matrix analysis. These metrics evaluate the discriminator's ability to differentiate between real and

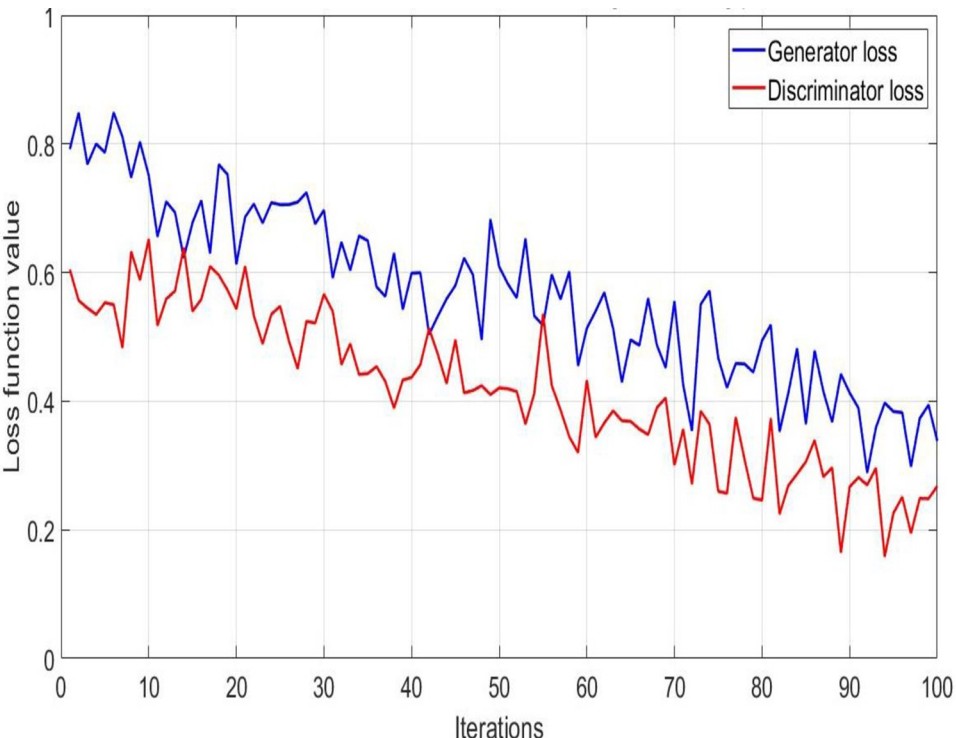

**Fig 6. Loss function variation curve during training process.**

generated training plans, offering deeper insights into the model's learning process and classification performance.

Table 4 provides a summary of the GAN discriminator's performance, with high accuracy (92.3%) and F1-Measure (91.1%), indicating its effectiveness in classifying real and generated plans. Balanced precision and recall scores demonstrate consistent reliability and sensitivity.

Fig 7 illustrates the GAN discriminator's predictions. True positives (450) and true negatives (485) dominate, with minimal false positives (35) and false negatives (30), reflecting the discriminator's strong classification ability.

**Analysis of the generation speed of personalized training plans.** To evaluate the efficiency of combining GAN to generate personalized motion training schemes, this experiment compared and analyzed the GAN model (Model A) with three traditional algorithm models: ML-based method (Model B), rule-based method (Model C), and statistical method (Model D). In this experiment, 10 athletes were randomly selected from the sample data, and the speed at which each model generated personalized sports training plans for each athlete was calculated. These data are presented in Table 5. The first column of Table 5 the athlete number and the data in columns 2 to 4 represent the time required to generate personalized sports

**Table 4. Discriminator performance metrics.**

| Metric | Value (%) |
|---|---|
| Accuracy | 92.3 |
| Precision | 91.5 |
| Recall | 90.7 |
| F1-Measure | 91.1 |

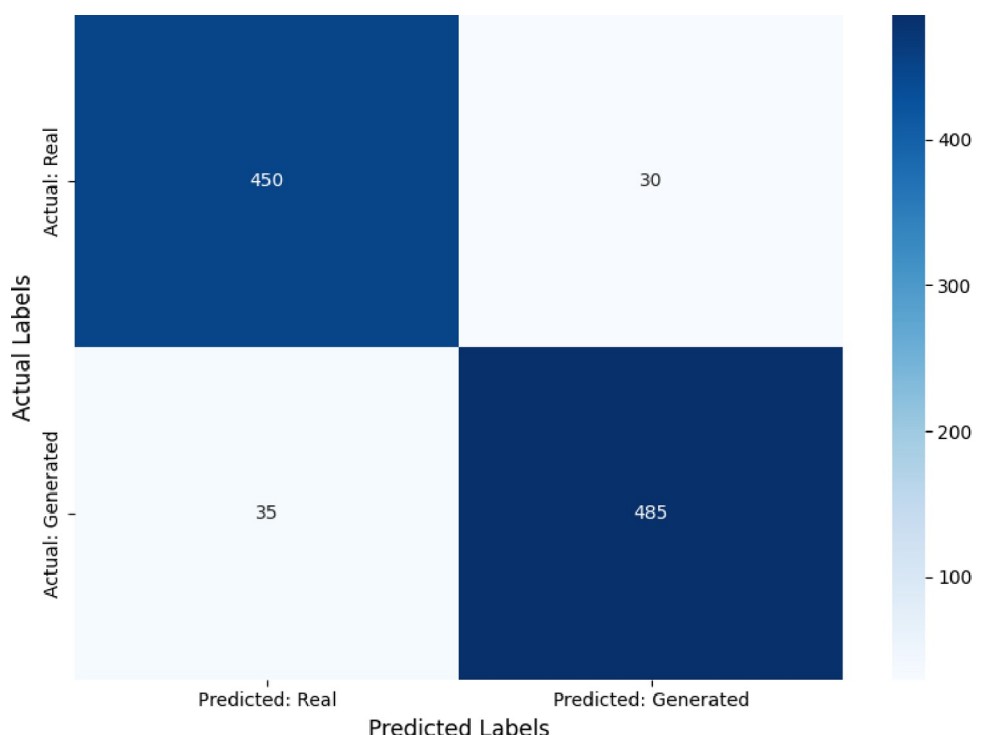

**Fig 7. Confusion matrix for GAN discriminator performance.**

training plans for each athlete under the four models. The data in Table 5 Under Model A (GAN model), each athlete has the shortest time to generate a plan, which means the generation speed is the fastest.

To observe more intuitively the efficiency of the GAN model compared to traditional algorithm models in generating personalized sports training plans, the average generation time of personalized plans for these 10 athletes under each model was calculated here. The calculation results are shown in Fig 8. For these 10 athletes, the average generation time of the GAN model (Model A) studied in this article is the shortest of 15.7 minutes. The average generation time of the traditional ML model (Model B) is 36.4 minutes; the average generation time of the

**Table 5. Record of generation speed for each model scheme.**

| Athlete number | Model A generation speed (minutes) | Model B generation speed (minutes) | Model C generation speed (minutes) | odel D generation speed (minutes) |
|---|---|---|---|---|
| 1 | 15 | 35 | 40 | 38 |
| 2 | 14 | 33 | 38 | 36 |
| 3 | 16 | 37 | 42 | 40 |
| 4 | 15 | 35 | 39 | 37 |
| 5 | 17 | 39 | 41 | 39 |
| 6 | 16 | 37 | 40 | 38 |
| 7 | 18 | 41 | 43 | 41 |
| 8 | 14 | 33 | 38 | 36 |
| 9 | 19 | 43 | 44 | 42 |
| 10 | 13 | 31 | 37 | 35 |

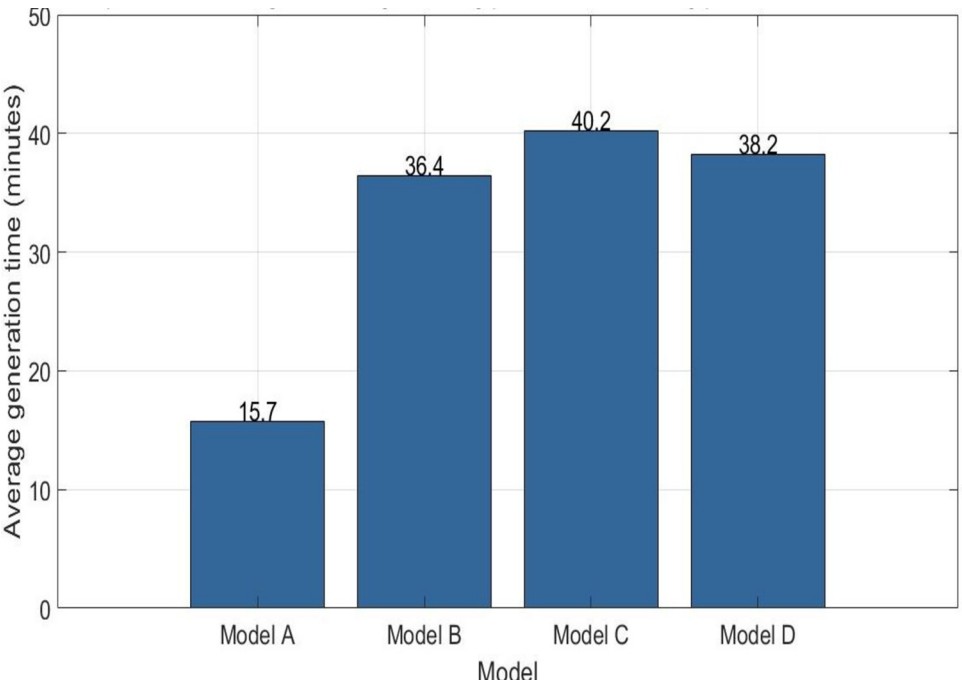

**Fig 8. Comparison of the average time for generating personalized training schemes for different models.**

rule-based method (Model C) is 40.2 minutes; the average generation time of the statistical method (Model D) is 38.2 minutes. The GAN model in this article has improved the efficiency of generating personalized motion training schemes by about 56.9% compared to traditional machine models. This indicates that the GAN model in this article has significant efficiency advantages in generating personalized sports training plans and can generate personalized sports training plans for athletes faster and more effectively.

**Comparative analysis of MSE between the model in this article and traditional model generation schemes.** Next, to explore the generation quality of personalized motion training schemes combined with GAN in this article, this experiment compared and analyzed the GAN model (Model A) with three traditional algorithm models: ML based method (Model B), rule-based method (Model C), and statistical method (Model D) on the MSE index. Among them, MSE is an indicator used to measure the average degree of difference between generated and real samples. A smaller MSE value means that the difference between the generated personalized motion training scheme and the real sample is smaller, indicating that the generator performs better in terms of generation quality. In this experiment, 10 athletes were randomly selected, and their MSE values for generating personalized sports training plans under different models were calculated. The calculation results are shown in Fig 9. In Fig 9, the horizontal axis represents the athlete number, and the vertical axis represents the MSE value. For these 10 athletes, the MSE of the GAN model in generating personalized sports training plans fluctuates between 0.08 and 0.12 in this article. The MSE values of the other three traditional algorithm models fluctuate between 0.13 and 0.22. Overall, the GAN model in this paper has a smaller MSE value than other traditional models. This indicates that the combination of the GAN model in this article can more accurately fit the needs and characteristics of athletes when generating personalized sports training plans. It can more effectively capture the feature distribution of samples and generate higher quality and more realistic personalized training plans.

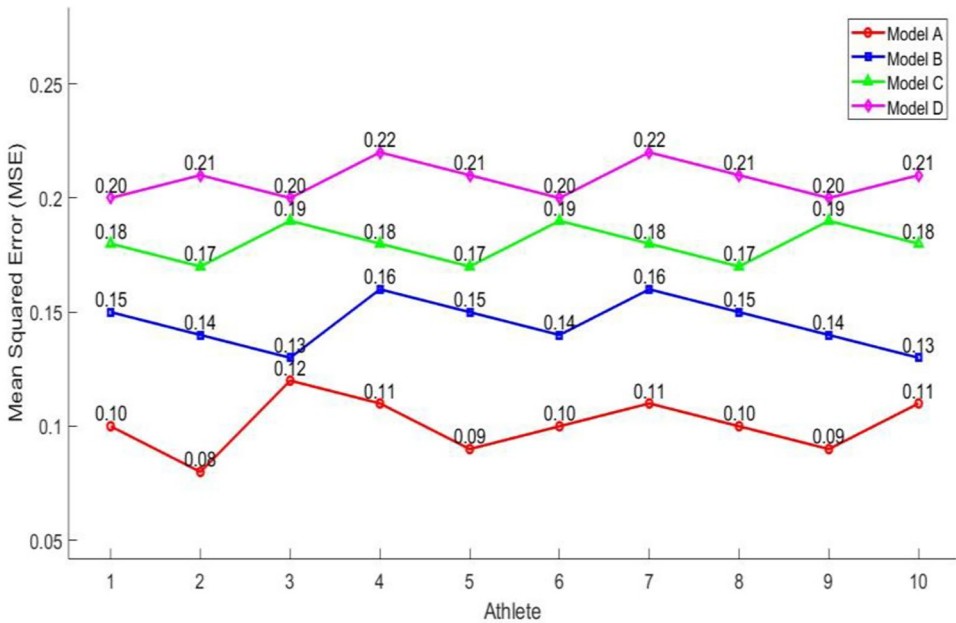

**Fig 9. Comparison of MSE for different models generating personalized schemes.**

While discussing the learning objective function, the aim to delve deeper into the mechanisms through which each model learns and optimizes its parameters. To achieve this, a series of ablation studies conducted to evaluate the influence of different input features and model components. The results reveal that integrating motion features significantly aligns the objective function with the desired outputs.

**Comparative analysis of real-time performance between the model in this article and the traditional model in generating personalized solutions.** To evaluate the real-time performance of combining GAN in generating personalized motion training schemes, this experiment compares the GAN model (Model A) with ML-based methods (Model B), rule-based methods (Model C), and statistical methods (Model D) through response time indicators. Ten athletes were selected for the experiment, and the response speed in generating personalized sports training plans for each athlete in each model was calculated. The calculation results are shown in Fig 10. From the data in Fig 10, it can be calculated that for these athletes, the average response time of the CAN model in generating personalized training plans in this article is 29.6ms. The average response times of traditional ML models and rule-based and rule-based methods are 50.8ms, 64.8ms, and 58.2ms, respectively. These data indicate that the GAN model in this article has the lowest response time compared to other traditional models. This indicates that the GAN model has good real-time performance in generating personalized training plans for athletes, providing real-time and efficient support for generating personalized sports training plans.

**The subjective evaluation of personalized schemes generated by this article's model and traditional ML models.** At the end of the experiment, a survey questionnaire was used to collect the evaluations of these 100 athletes on the personalized plans generated by the GAN model and traditional ML models. The evaluation content includes scoring the context-specific, scientificity, applicability, and feasibility of the generated plan in four aspects, with a score of ten. This experiment collected and calculated the average scores of these 100 athletes and recorded the results in Table 6. The GAN model in this article has achieved higher ratings

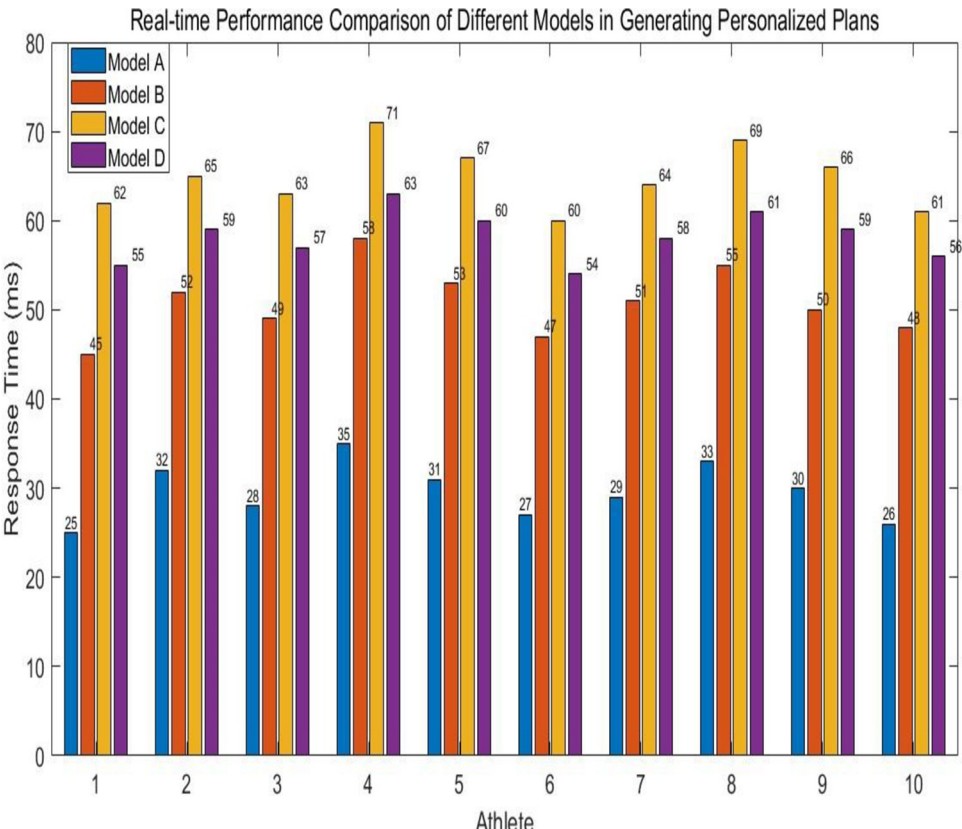

**Fig 10. Real time comparison of personalized schemes generated by different models.**

than traditional ML models in four aspects of evaluation. This indicates that combining GAN's personalized sports training plan is relatively reasonable. A training plan that meets athletes' needs is more suitable for them to train and is conducive to improving their training effectiveness.

The proposed GAN-based model is compared to existing state-of-the-art methods across key metrics, including generation speed, MSE, context-specific, and applicability scores, as shown in Table 7.

The proposed model outperforms traditional ML and rule-based approaches by efficiently integrating numeric and motion data, resulting in superior context-specific and lower MSE. While conditional GANs and IoT-based methods perform competitively, the proposed approach has more flexibility and scalability. Traditional methods, such as random forest and gradient-boosted trees, offer limited adaptability and struggle with multimodal data fusion, highlighting the advantages of the GAN model in generating high-quality, context-specific training plans.

**Table 6. Evaluation of personalized training schemes generated by different models.**

| Model | Context-specific Level | Scientificity | Applicability | Feasibility |
|---|---|---|---|---|
| GAN Model | 9.32 | 9.00 | 8.57 | 8.80 |
| Traditional ML Model | 7.52 | 7.70 | 7.25 | 7.44 |

**Table 7. Comparison with existing state-of-the-art methods.**

| Method | Reference | Generation Speed (min) | MSE | Context-specific Score | Applicability Score |
|---|---|---|---|---|---|
| Proposed GAN Model | Proposed study | 15.7 | 0.035 | 4.8/5 | 4.7/5 |
| Conditional GAN (CGAN) | Wu et al. [16] | 18.3 | 0.045 | 4.5/5 | 4.3/5 |
| Random Forest-Based Model | Cao et al. [12] | 36.2 | 0.070 | 3.8/5 | 3.9/5 |
| Gradient Boosted Trees | Shin et al. [11] | 32.5 | 0.062 | 4.0/5 | 4.1/5 |
| Rule-Based System | Pickering and Kiely [28] | 45.1 | 0.092 | 3.5/5 | 3.4/5 |
| IoT-AI Integrated Framework | Li and Shi [13] | 27.5 | 0.055 | 4.2/5 | 4.0/5 |

## 5. Conclusions

This study presents a novel approach for generating context-specific sports training plans using GAN technology. The proposed model integrates numeric attributes (e.g., age, heart rate) and motion features extracted from video data, offering a unique multimodal framework that enhances context-specific and adaptability. By leveraging adversarial training, the model dynamically improves the quality and relevance of generated plans, addressing athletes' diverse and evolving needs. Experimental results demonstrate that the GAN-based model performs better than traditional ML and rule-based methods, delivering higher efficiency, reduced MSE, and improved real-time adaptability. Subjective evaluations involving 30 respondents, including athletes and professional coaches, further validate the model's effectiveness. Using a five-point Likert scale, participants rated the generated plans on context-specific, scientificity, applicability, and feasibility, with the GAN model significantly outperforming traditional methods in all dimensions. Statistical significance was established through one-way ANOVA and post hoc tests, reinforcing the model's contributions. However, the study acknowledges certain limitations. The data collection process, while comprehensive, may be constrained by variability in data quality and diversity, potentially impacting model robustness. The GAN training process also requires substantial computational resources and time, which may limit scalability in resource-constrained environments. Future research will focus on optimizing data collection techniques to improve data quality and representativeness, exploring more efficient model architectures and training algorithms to reduce computational demands, and incorporating domain expertise to enhance the practical applicability of training plans further. These findings highlight the transformative potential of GAN-based methods in advancing personalized sports training methodologies, with implications for scalable, adaptive, and high-quality plan generation in real-world athletic environments.

## Author Contributions

**Conceptualization:** Juquan Tan.

**Methodology:** Jingwen Chen.

**Writing – original draft:** Juquan Tan.

**Writing – review & editing:** Juquan Tan.

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
