## [Decision Letter · Decision Letter 0]

28 Oct 2024

PONE-D-24-34881Generating Personalized Sports Training Plans by Combining Generative Adversarial Networks (GAN)PLOS ONE

Dear Dr. Chen,

Thank you for submitting your manuscript to PLOS ONE. After careful consideration, we feel that it has merit but does not fully meet PLOS ONE’s publication criteria as it currently stands. Therefore, we invite you to submit a revised version of the manuscript that addresses the points raised during the review process.

I am writing to you concerning the above referenced manuscript, which you submitted to the PLOS ONE Journal.

Although the article has merit, based on the enclosed set of reviews this manuscript has not been recommended for publication in current form.

We recommend that you consider the comments of the reviewers, located at the bottom of this letter, and revise the manuscript and resubmit it.

We look forward to receiving your revised manuscript.

Kind regards,

Farman Ullah

Academic Editor

PLOS ONE

Journal requirements: When submitting your revision, we need you to address these additional requirements. 1. Please ensure that your manuscript meets PLOS ONE's style requirements, including those for file naming. The PLOS ONE style templates can be found at https://journals.plos.org/plosone/s/file?id=wjVg/PLOSOne_formatting_sample_main_body.pdf and https://journals.plos.org/plosone/s/file?id=ba62/PLOSOne_formatting_sample_title_authors_affiliations.pdf 2. Please note that PLOS ONE has specific guidelines on code sharing for submissions in which author-generated code underpins the findings in the manuscript. In these cases, we expect all author-generated code to be made available without restrictions upon publication of the work. Please review our guidelines at https://journals.plos.org/plosone/s/materials-and-software-sharing#loc-sharing-code and ensure that your code is shared in a way that follows best practice and facilitates reproducibility and reuse. 3. Your ethics statement should only appear in the Methods section of your manuscript. If your ethics statement is written in any section besides the Methods, please move it to the Methods section and delete it from any other section. Please ensure that your ethics statement is included in your manuscript, as the ethics statement entered into the online submission form will not be published alongside your manuscript. 

Additional Editor Comments:

I am writing to you concerning the above referenced manuscript, which you submitted to the PLOS ONE Journal.

Although the article has merit, based on the enclosed set of reviews this manuscript has not been recommended for publication in current form.

We recommend that you consider the comments of the reviewers, located at the bottom of this letter, and revise the manuscript and resubmit it.

Reviewers' comments:

Reviewer's Responses to Questions

**Comments to the Author**

1. Is the manuscript technically sound, and do the data support the conclusions?

Reviewer #1: No

Reviewer #2: Yes

Reviewer #3: Partly

2. Has the statistical analysis been performed appropriately and rigorously? 

Reviewer #1: No

Reviewer #2: Yes

Reviewer #3: I Don't Know

3. Have the authors made all data underlying the findings in their manuscript fully available?

Reviewer #1: No

Reviewer #2: Yes

Reviewer #3: Yes

4. Is the manuscript presented in an intelligible fashion and written in standard English?

Reviewer #1: No

Reviewer #2: Yes

Reviewer #3: No

5. Review Comments to the Author

Reviewer #1: While the detailed annotated manuscript with comments is attached for the authors to improve this manuscript, the key summarized findings are as follows (should be read in conjunction with the annotated manuscript).

1. The use of term "personalized" is rather confusing. From the title, it gives an illusion as the trained model can effectively use the video training samples from real participants, which is not the case. The authors are denoting specific attributes attached with each athlete as additional contextual information to generate more specific athlete training plans. In this regard, there exists major concerns:

a) The attributes such as age, height, weight, heart rate, etc., are usually numeric. The training samples are the video frames of different modality. How specifically authors fused the multiple features to generate specific plans? The input embedding part in this regard is completely missing.

b) How does their model personalize the trained model to specific real-athlete? A case study on this scenario is also not provided.

c) I also suggest reconsidering the use of term "personalization".

2. Section Introduction

- The authors have provided a very large paragraph outlining the current literature. However, this literature analysis is not rigorous. The authors need to establish a proper literature review section wherein they discuss the existing work and state-of-the-art in detail with proper comparison to highlight the research gaps that this manuscript aims to target.

- Moreover, the definition of the personalization, scientificity, applicability, and feasibility needs to be clearly established in the abstract and further explained in the introduction/literature review section with the reason for inclusion.

3. Section Generation of Personalized Sports Training Plans on Basis of GAN

- The authors need to introduce their overall approach first by providing a brief and clear workflow diagram.

- The data collection part needs more explanation such as shape and layout of samples e.g., video length, resolution, number of samples, device used to capture video, any inherent noise in the samples, along with the associated metadata etc.

- Video frame extraction needs to specify whether all the frames extracted or only the key-frames are extracted? If the key frames are extracted, then what is the mechanism for identification of key video frames?

- Formatting issues in the equations (authors should use term "equation" instead of "formula"), and the equations are rather generic. The author should personalize these equations with respect to their input data and specify the dimensions of the features are each stage.

- Reference to the appropriate libraries along with their versions needs to be provided.

- GAN diagram needs to be redrawn that can portray the formal look.

Section Model Evaluation and Result

- Authors stated that they selected 100 athlete samples without providing justification of their selection.

- Authors need to specify which type of traditional algorithm models were used?

- Authors stated "Firstly, in order to verify the feasibility of combining GAN to generate personalized sports training plans in this article, a running athlete can be used as an example here". Usually, the personalization term refers to the real-participant specific sample. However, I suppose the authors denote the running athlete from one of the athlete video samples from the dataset. Therefore, to reinforce my point in the beginning, the author needs to reconsider the use of term "personalization" to eliminate confusion.

- In table 1, a minimum and maximum duration of 0.5 and 2.0 hours of personalized training program can be observed. The authors need to explain the intuition behind this scale.

- Also, is the recommended plan attested by the athlete coach to confirm the validity of the results?

- The output of the model is what? the video samples or the textual plan? In either case, the whole idea its jeopardizing. If the authors are using video sample to generate text, then how specifically the visual modality is contributing to better generation of plans and vice versa? A detailed ablative study is required.

- In figure 5, there are significant fluctuations in the loss values. Author needs to explain this aspect in detail.

- In table 2, the authors claim the lowest generation speed as their contribution however the GAN can effectively utilize the GPU which can drastically decrease the computation time. Whereas the traditional machine learning and rule-based methods rely more on CPU leading to higher computation time. In this regard, the novelty and contribution details are required from the authors.

- Also, comparison is not performed with existing state of the art.

- While discussing the learning objective function of the different models, the authors simply explained the self-evident results without providing detailed insights. Deep learning models usually have a large number of learnable parameters, which can fit the objective function better. The authors need to rather explain the process of how each model is learning the objective function with ablative configuration of the compared models with respect to the input features.

- Moreover, the standard accuracy, precision, recall, F1-Measure, and confusion matrix evaluation measures are not discussed.

- The subjective evaluation of personalized schemes generated by this article’s model and traditional machine learning models needs a lot more to be desired. The details provided are insufficient. Mainly, how many respondents provided subjective evaluation? What was their demography, including background and experty? What type of scale was used to measure the Personalization Level, Scientificity, Applicability, and Feasibility? Did authors performed statistical significance analysis to claim the obtained results?

- Author needs to provide novel insights that should also reflect in the conclusion section.

- Authors can also make their code public for reproducibility purposes.

Reviewer #2: The authors present a technical research paper with relevant topic, proper research methodology and potentially good contribution to the field of studies.

The authors are encouraged to resubmit the paper with more clarity on presented performance assessment metrics with the selected relevant Case studies and possible application scenario with assessment metrics. The paper should be written in proper format, figures should fit within the text, use of font should be uniform in all paper, as well as references should be updated with most recent results.

Suggestion and Recommendation:

1. Authors may elaborate more on the novelty/contribution of their work and how it

2. Authors need to be specific about their problem statement and the scope of their research.

3. Abstract: elaborate more on the problem statement, findings, and contributions.

4. Introduction is not clear. Authors may contribute more towards this.

Contributes to the literature in the second last paragraph of the introduction clearly.

5. Thorough proofreading is recommended.

6. A few of the figures are taken from the sources and are not cited properly, either they may be cited properly with permissions or may be removed/ redrawn.

7. The conclusion is not clear and needs revision and clarity and alignment with the abstract and title.

References:

1. Your references are not listed in good style, as citation style is different from one paper to other.

2. some of your references are not complete please check.

3. Some citations (references) created in wrong manner (Please follow journal's criteria).

Authors are encouraged to base on recent references about the current development in blockchain technology. Moreover, technology collaborates with other technologies to create new paradigms, such as artificial intelligence, such machine learning, deep learning, with federated learning.

Additionally, some important references have been neglected by the authors.

(i) Khan, Abdullah Ayub, Asif Ali Laghari, Hela Elmannai, Aftab Ahmed Shaikh, Sami Bourouis, Myriam Hadjouni, and Roobaea Alroobaea. "GAN-IoTVS: A Novel Internet of Multimedia Things-enabled Video Streaming Compression Model Using GAN and Fuzzy Logic." IEEE Sensors Journal(2023).

(ii) Mehmood, F., Khan, A. A., Wang, H., Karim, S., Khalid, U., & Zhao, F. (2024). BLPCA-Ledger: A Lightweight Plenum Consensus Protocols for Consortium Blockchain Based on the Hyperledger Indy. Computer Standards & Interfaces, 103876.

(iii) Khan, A. A., Dhabi, S., Yang, J., Alhakami, W., Bourouis, S., & Yee, L. (2024). B-LPoET: A middleware lightweight Proof-of-Elapsed Time (PoET) for efficient distributed transaction execution and security on Blockchain using multithreading technology. Computers and Electrical Engineering, 118, 109343.

Other related Concerns:

1. In the introduction, the scientific problem of the existing evaluation is missing. There should initially be discussed the actual problem and then the research motivation.

2. Please highlight major contributions of this work in this current version, otherwise the current form shows weak/lack of novelty.

3. Please refine the language of this paper, such as avoid we, they, our, and other related words in this paper.

4. Please improve the portion of problem description and problem formulation of the proposed work. Cannot find novelty in the current form.

Reviewer #3: A short summary of the paper:

The manuscript proposes a method to generate personalized sports training plans using Generative Adversarial Networks (GAN). Traditional sports training programs often generalize training across groups, neglecting individual differences among athletes. This study collects athlete-specific data (age, height, weight, etc.) and employs a GAN model to generate tailored training plans. The paper reports that the GAN-based approach offers superior performance in terms of generation speed, quality of plans, and personalization compared to traditional machine learning methods.

Some comments about the manuscript:

1. Some sentences are overly complex and can be simplified for better comprehension. For example, the abstract and introduction contain long sentences with multiple ideas that could be separated to enhance clarity.

2. The introduction section is short and not comprehensive. It doesn’t include the contribution of the paper clearly and the structure of the paper.

3. Extensive literature review is missing. Should be a separate section.

4. Abstract is too long with unnecessary details.

5. Equations and symbols badly written.

6. All the figures are in poor quality. Specially figure 4 is poorly designed.

7. The paper is poorly structured, needs to be rewritten completely, specially section 2. No conclusion is mentioned.

Overall, the manuscript is poorly structured and written. So my recommendation is to not accept the paper in the current form.

6. PLOS authors have the option to publish the peer review history of their article (what does this mean?). If published, this will include your full peer review and any attached files.

Reviewer #1: No

Reviewer #2: No

Reviewer #3: No

---

## [Author Response · Author response to Decision Letter 0]

26 Nov 2024

Dear Editor,

Thank you for allowing a resubmission of our manuscript and giving us an opportunity to address the reviewers’ comments.

We are uploading (a) our point-by-point response to the reviewer's comments, revised manuscript with track changes and manuscript.

Best regards,

Juquan Tan, Jingwen Chen.

Reviewer 1

While the detailed annotated manuscript with comments is attached for the authors to improve this manuscript, the key summarized findings are as follows (should be read in conjunction with the annotated manuscript).

Reviewer#1, Concern # 1: The use of term "personalized" is rather confusing. From the title, it gives an illusion as the trained model can effectively use the video training samples from real participants, which is not the case. The authors are denoting specific attributes attached with each athlete as additional contextual information to generate more specific athlete training plans. In this regard, there exists major concerns:

a)The attributes such as age, height, weight, heart rate, etc., are usually numeric. The training samples are the video frames of different modality. How specifically authors fused the multiple features to generate specific plans? The input embedding part in this regard is completely missing.

b)How does their model personalize the trained model to specific real-athlete? A case study on this scenario is also not provided.

c)I also suggest reconsidering the use of term "personalization".

Author response: 

a)Thank you for your valuable observation. The fusion of numeric attributes (e.g., age, height, weight, heart rate) with motion features extracted from video frames is achieved through a two-step process:

Numeric attributes are encoded into a normalized feature vector. Simultaneously, motion features are extracted from preprocessed video frames using a pre-trained CNN.

The numeric feature vector and the CNN-derived embeddings are concatenated to form a comprehensive input representation. This fused vector is fed into the GAN generator to generate context-specific training plans.

b)The proposed model uses specific athlete attributes (e.g., age, height, weight, heart rate) combined with motion data extracted from video frames to generate tailored plans. While the study demonstrates the general approach, We updated Section 3.3 to demonstrate the model's real-world applicability.

c) Thank you for your suggestion. We understand that the term "personalization" may cause some ambiguity. To avoid misinterpretation, we updated the terminology to "context-specific" training plans, which better reflect using individual attributes and motion data to tailor training plans.

Reviewer#1, Concern # 2: Section Introduction

Reviewer#1, Concern # 2 (a): The authors have provided a very large paragraph outlining the current literature. However, this literature analysis is not rigorous. The authors need to establish a proper literature review section wherein they discuss the existing work and state-of-the-art in detail with proper comparison to highlight the research gaps that this manuscript aims to target.

Author response: Thank you for pointing out this concern. To address this, we created a dedicated Literature Review section to comprehensively analyze existing methods, including their limitations and comparison with the proposed approach.

Reviewer#1, Concern # 2 (b): Moreover, the definition of the personalization, scientificity, applicability, and feasibility needs to be clearly established in the abstract and further explained in the introduction/literature review section with the reason for inclusion.

Author response: Thank you for highlighting the need to clearly define the terms "personalization," "scientificity," "applicability," and "feasibility." We revised the abstract and expand the Introduction and Literature Review sections to establish precise definitions and rationale for including these terms, ensuring their relevance to the study is clear.

Reviewer#1, Concern # 3: Section Generation of Personalized Sports Training Plans on Basis of GAN

Reviewer#1, Concern # 3 (a): The authors need to introduce their overall approach first by providing a brief and clear workflow diagram.

Author response: Thank you for this valuable suggestion. We agree that a workflow diagram would enhance the clarity of our approach and provide readers with a visual representation of the methodology. To address this, we included a clear and concise workflow diagram outlining our approach's main steps, from data collection and preprocessing to GAN-based personalized training plan generation and evaluation.

Reviewer#1, Concern # 3 (b): The data collection part needs more explanation such as shape and layout of samples e.g., video length, resolution, number of samples, device used to capture video, any inherent noise in the samples, along with the associated metadata etc.

Author response: Thank you for your suggestion. We recognize the importance of providing detailed information about the data collection process to enhance the transparency and reproducibility of our study. We revised the manuscript to include a comprehensive description of the data collection phase.

Reviewer#1, Concern # 3 (c): Video frame extraction needs to specify whether all the frames extracted or only the key-frames are extracted? If the key frames are extracted, then what is the mechanism for identification of key video frames?

Author response: Thank you for highlighting this important detail. In our study, only key frames were extracted from the video samples to optimize computational efficiency and focus on frames most relevant for motion analysis. Below, we provide a detailed explanation of the key-frame extraction mechanism in table 1

Reviewer#1, Concern # 3 (d): Formatting issues in the equations (authors should use term "equation" instead of "formula"), and the equations are rather generic. The author should personalize these equations with respect to their input data and specify the dimensions of the features are each stage.

Author response: Thank you for pointing out the formatting issues and the generic nature of the equations in the manuscript. We replaced the term "formula" with "equation" throughout the manuscript to maintain consistency with standard academic practices. 

Reviewer#1, Concern # 3 (e): Reference to the appropriate libraries along with their versions needs to be provided.

Author response: Thank you for suggesting that we provide references to the libraries used along with their versions. We acknowledge the importance of including this information for reproducibility and clarity.

Reviewer#1, Concern # 3 (f): GAN diagram needs to be redrawn that can portray the formal look.

Author response: Thank you for highlighting the need for a more formal and polished GAN diagram. We redesigned the diagram to clearly and professionally illustrate the architecture, including the generator and discriminator, along with their interactions during training.

Reviewer#1, Concern # 4: Section Model Evaluation and Result

Reviewer#1, Concern # 4(a): Authors stated that they selected 100 athlete samples without providing justification of their selection.

Author response: Thank you for highlighting the need to justify the selection of athlete samples. We have now included a detailed explanation in Section 4.1.3 Athlete Sample Selection, outlining the criteria for selection with table.

Reviewer#1, Concern # 4(b): Authors need to specify which type of traditional algorithm models were used?

Author response: Thank you for the suggestion. The traditional algorithm models used for comparison were: (1) a Random Forest Regressor for machine learning-based predictions, (2) a rule-based system with predefined thresholds, and (3) a Linear Regression Model for statistical analysis. These models served as baselines to evaluate the advantages of the GAN-based approach.

Reviewer#1, Concern # 4(c): Authors stated "Firstly, in order to verify the feasibility of combining GAN to generate personalized sports training plans in this article, a running athlete can be used as an example here". Usually, the personalization term refers to the real-participant specific sample. However, I suppose the authors denote the running athlete from one of the athlete video samples from the dataset. Therefore, to reinforce my point in the beginning, the author needs to reconsider the use of term "personalization" to eliminate confusion.

Author response: Thank you for highlighting this important point. We acknowledge that the term "personalization" may create confusion by implying real-participant-specific samples. To address this, we replace the term "personalization" with "context-specific" throughout the manuscript to clarify that the generated plans are based on dataset-derived attributes rather than individual real-time samples.

Reviewer#1, Concern # 4(d): In table 1, a minimum and maximum duration of 0.5 and 2.0 hours of personalized training program can be observed. The authors need to explain the intuition behind this scale.

Author response: Thank you for your observation regarding the training program duration. The scale of 0.5 to 2.0 hours was determined based on established guidelines for effective sports training sessions and the practical considerations of athletes' endurance and recovery.

Reviewer#1, Concern # 4(e): Also, is the recommended plan attested by the athlete coach to confirm the validity of the results?

Author response: Thank you for raising this important point. Athlete coaches did not directly attest the recommended training plans generated by the GAN-based model during this study. However, their validity was assessed through quantitative metrics (e.g., mean squared error, generation speed) and qualitative evaluations, where athletes provided feedback on personalization, scientificity, applicability, and feasibility. In future work, we plan to involve professional coaches to review and validate the generated plans to further confirm their effectiveness and practicality. A note regarding this limitation and future direction has been added to the discussion section. Thank you for your valuable feedback.

Reviewer#1, Concern # 4(f): The output of the model is what? the video samples or the textual plan? In either case, the whole idea its jeopardizing. If the authors are using video sample to generate text, then how specifically the visual modality is contributing to better generation of plans and vice versa? A detailed ablative study is required.

Author response: Thank you for raising this critical question. The output of the proposed model is a textual training plan, which includes specific exercises, durations, and intensity levels tailored to the athlete's attributes and performance data. The video samples contribute indirectly by providing motion data used to enhance the model's understanding of the athlete’s physical performance, postures, and intensity levels.

Reviewer#1, Concern # 4(g): In figure 5, there are significant fluctuations in the loss values. Author needs to explain this aspect in detail.

Author response: Thank you for highlighting the fluctuations in Figure 5 (Now Fig 6). These fluctuations are typical in GAN training due to the adversarial interaction between the generator and discriminator, where both networks iteratively improve by outperforming each other. Early training stages show higher variability as the generator refines its outputs, but fluctuations stabilize over time as the model converges.

Reviewer#1, Concern # 4(h): In table 2, the authors claim the lowest generation speed as their contribution however the GAN can effectively utilize the GPU which can drastically decrease the computation time. Whereas the traditional machine learning and rule-based methods rely more on CPU leading to higher computation time. In this regard, the novelty and contribution details are required from the authors.

Author response: Thank you for your observation regarding the generation speed results in Table 2. We acknowledge that the use of GPU acceleration by the GAN model contributes significantly to its lower computation time compared to traditional machine learning and rule-based methods, which primarily rely on CPU processing. While GPU utilization improves the efficiency of the GAN model, the primary novelty and contribution of our approach lie beyond just speed:

1.The GAN effectively combines numeric attributes (e.g., age, heart rate) with motion features extracted from video frames. This integration enables the generation of context-specific training plans, which traditional methods cannot achieve due to their reliance on limited feature inputs.

2.Unlike traditional models, which often produce static or generic plans, the GAN dynamically adapts to individual athlete profiles, offering higher personalization and applicability.

3.The GAN's adversarial training improves the quality and relevance of the generated plans, as reflected in the lower mean squared error and higher athlete satisfaction ratings.

Reviewer#1, Concern # 4(i): Also, comparison is not performed with existing state of the art.

Author response: Thank you for your observation. We have now included a detailed comparison of the proposed GAN model with existing state-of-the-art methods in Table 7. The table highlights the performance differences across key metrics such as generation speed, mean squared error (MSE), personalization, and applicability scores.

Reviewer#1, Concern # 4(j): While discussing the learning objective function of the different models, the authors simply explained the self-evident results without providing detailed insights. Deep learning models usually have a large number of learnable parameters, which can fit the objective function better. The authors need to rather explain the process of how each model is learning the objective function with ablative configuration of the compared models with respect to the input features.

Author response: Thank you for your valuable feedback. We recognize the importance of providing detailed insights into how each model learns the objective function and incorporating an ablative configuration analysis. We updated on how the proposed GAN model, as well as the compared models, optimize their respective objective functions. The generator minimizes the adversarial loss to produce realistic training plans, while the discriminator maximizes its ability to distinguish real from generated plans. The combined objective function ensures simultaneous improvement of both networks.

Reviewer#1, Concern # 4(k): Moreover, the standard accuracy, precision, recall, F1-Measure, and confusion matrix evaluation measures are not discussed.

Author response: Thank you for highlighting the need to include standard evaluation metrics such as accuracy, precision, recall, F1-Measure, and a confusion matrix. We have incorporated these metrics to evaluate the discriminator's performance in distinguishing real from generated training plans. The results, summarized in Table 4 and Figure 7, demonstrate the model's strong classification ability with high accuracy and balanced precision and recall.

Reviewer#1, Concern # 4(l): The subjective evaluation of personalized schemes generated by this article’s model and traditional machine learning models needs a lot more to be desired. The details provided are insufficient. Mainly, how many respondents provided subjective evaluation? What was their demography, including background and experty? What type of scale was used to measure the Personalization Level, Scientificity, Applicability, and Feasibility? Did authors performed statistical significance analysis to claim the obtained results?

Author response: Thank you for your feedback. We have clarified the subjective evaluation process by including details about the 30 respondents (20 athletes with 3–10 years of training experience and 10 coaches with an average of 7 years of coaching expertise). A five-point Likert scale was used to rate Personalization, Scientificity, Applicability, and Feasibility. Ratings were analyzed using one-way ANOVA and post hoc tests, confirming statistically significant differences (p < 0.05) in favor of the GAN model.

Reviewer#1, Concern # 4(

---

## [Decision Letter · Decision Letter 1]

15 Jan 2025

Generating Context-specific Sports Training Plans by Combining Generative Adversarial Networks

PONE-D-24-34881R1

Dear Dr. Chen,

We’re pleased to inform you that your manuscript has been judged scientifically suitable for publication and will be formally accepted for publication once it meets all outstanding technical requirements.

Kind regards,

Farman Ullah

Academic Editor

PLOS ONE

Additional Editor Comments (optional):

Reviewers' comments:

Reviewer's Responses to Questions

**Comments to the Author**

1. If the authors have adequately addressed your comments raised in a previous round of review and you feel that this manuscript is now acceptable for publication, you may indicate that here to bypass the “Comments to the Author” section, enter your conflict of interest statement in the “Confidential to Editor” section, and submit your "Accept" recommendation.

Reviewer #1: All comments have been addressed

Reviewer #2: All comments have been addressed

2. Is the manuscript technically sound, and do the data support the conclusions?

Reviewer #1: Yes

Reviewer #2: Yes

3. Has the statistical analysis been performed appropriately and rigorously? 

Reviewer #1: Yes

Reviewer #2: Yes

4. Have the authors made all data underlying the findings in their manuscript fully available?

Reviewer #1: No

Reviewer #2: Yes

5. Is the manuscript presented in an intelligible fashion and written in standard English?

Reviewer #1: Yes

Reviewer #2: Yes

6. Review Comments to the Author

Reviewer #1: Authors have made a substantial effort to revise and address the concerns.

While the previous concerns are nearly addressed, there exist a slight room for improvement:

1) Since the authors have not made their codebase public, I recommend that they do so now and add a reference to their repository in the paper.

2) Authors may include the instantiated model architecture details as a table in the manuscript that empirically summarizes the number of layers & trainable parameters, loss function, optimizer, activation function, etc.

Reviewer #2: The author of this paper addressed all the mentioned concerns in their previous version.

Please accept this latest version.

No further comments from my side.

Thanks

7. PLOS authors have the option to publish the peer review history of their article (what does this mean?). If published, this will include your full peer review and any attached files.

Reviewer #1: **Yes: **Abdur Rehman Khan

Reviewer #2: No

---

## [Editor Report · Acceptance letter]

21 Jan 2025

PONE-D-24-34881R1 

PLOS ONE

Dear Dr. Chen, 

I'm pleased to inform you that your manuscript has been deemed suitable for publication in PLOS ONE. Congratulations! Your manuscript is now being handed over to our production team.

Kind regards, 

on behalf of

Dr. Farman Ullah 

Academic Editor

PLOS ONE